# Tuning the zeolite acidity enables selectivity control by suppressing ketene formation in lignin catalytic pyrolysis

Zeyou Pan [1,2], Allen Puente-Urbina [2,3], Syeda Rabia Batool[2], Andras Bodi [1], Xiangkun Wu[1], Zihao Zhang[1], Jeroen A. van Bokhoven [1,2] ✉ & Patrick Hemberger [1] ✉

Unveiling catalytic mechanisms at a molecular level aids rational catalyst design and selectivity control for process optimization. In this study, we find that the Brønsted acid site density of the zeolite catalyst efficiently controls the guaiacol catalytic pyrolysis mechanism. Guaiacol demethylation to catechol initiates the reaction, as evidenced by the detected methyl radicals. The mechanism branches to form either fulvenone ($c\text{-}C_5H_4 = C = O$), a reactive ketene intermediate, by catechol dehydration, or phenol by acid-catalyzed dehydroxylation. At high Brønsted acid site density, fulvenone formation is inhibited due to surface coordination configuration of its precursor, catechol. By quantifying reactive intermediates and products utilizing operando photoelectron photoion coincidence spectroscopy, we find evidence that ketene suppression is responsible for the fivefold phenol selectivity increase. Complementary fulvenone reaction pathway calculations, along with [29]Si NMR-MAS spectroscopy results corroborate the mechanism. The proposed, flexible operando approach is applicable to a broad variety of heterogeneous catalytic reactions.

Rational catalyst design for selectivity control is the Holy Grail in heterogeneous catalysis[1]. It is made possible by understanding the mechanism on the catalyst surface as well as in the gas phase. Microkinetic modeling is an important tool to identify critical intermediates and rate-limiting elementary reactions in all phases and states of the catalytic process[2]. This information is fed back into the catalyst design loop to optimize the selectivity. Reactive intermediates, such as ketene (ethenone, $CH_2CO$) in the Fischer–Tropsch[3–5] and carbenes in the zeolite-catalyzed methanol-to-hydrocarbons (MTH) process[6], correspond to local minima on the reaction potential energy surface and are key to driving product branching and, thus, determining selectivities[6,7]. In the MTH reaction, ketene is stabilized as surface acetal or a surface-bound acylium ion, as identified computationally and by IR spectroscopy[8,9]. Recently, Chen et al.[10] discussed the surface

confinement effect on the formation, fate, and catalytic reactivity of ketenes as a function of the channel size in mordenite (MOR) catalysts. However, ketenes are short-lived and must be detected under operando conditions to understand their role in the catalytic mechanism. Such reactive intermediates desorbing from the catalyst can be detected in the gas phase using photoionization mass spectrometry and photoelectron photoion coincidence (PEPICO) techniques with synchrotron radiation. PEPICO provided mechanistic insights into MTH[11–13], C–H activation in alkanes[14,15], and lignin pyrolysis[16–22].

In this contribution, we show how unveiling mechanistic details can help to steer the reactive flux towards a preferred final product. The strategies introduced here may guide rational catalyst design to induce a mechanism change in other catalytic systems, too. This proof-of-principle work addresses the optimization of the zeolite-catalyzed

[1]Paul Scherrer Institute, Forschungsstrasse 111, CH-5232 Villigen PSI, Switzerland. [2]Institute for Chemical and Bioengineering, Department of Chemistry and Applied Biosciences, ETH Zurich, 8093 Zurich, Switzerland. [3]Present address: Renewable Resources and Enabling Sciences Center, National Renewable Energy Laboratory, Golden, CO 80401, USA. ✉e-mail: jeroen.vanbokhoven@chem.ethz.ch; patrick.hemberger@psi.ch

**Fig. 1 | Reaction mechanism of guaiacol catalytic fast pyrolysis.** Hydrogen addition/subtraction and methylation pathways are symbolized as free radical reactions but likely take place preferentially on the zeolite catalyst surface.

fast pyrolysis (CFP) of guaiacol (methoxyphenol), a lignin model compound, to yield preferentially phenol. Guaiacol **1** CFP is initiated by demethylation and hydrogen transfer to yield catechol **2** (*ortho*-benzenediol, **1→2**, Fig. 1) and surface methyl or a methyl radical[20,23,24]. Catechol **2** undergoes intramolecular dehydration to afford fulvenone ketene **4**, in line with recent highlights and reviews emphasizing the key role of ketenes in zeolite chemistry[6,7,11,12]. Catechol dehydroxylation (**2→3**) to phenol (*m/z* 94) and decarbonylation (**2→5**) to 4-cyclopenten-1-one (*m/z* 82) also contribute to the overall reactivity. The fulvenone ketene channel leads to branching of the reactive flux and a broad product distribution, including cyclopentadiene **6**, methyl cyclopentadiene **7**, fulvene **8**, benzene **9**, and phenol **3** (Fig. 1)[18,19]. It is, therefore, reasonable to expect that phenol selectivity can be increased targetedly by suppressing fulvenone ketene formation.

Compared to Lewis acid sites, Brønsted acid sites represent the main active sites in catalytic pyrolysis over zeolites[25–27]. In our previous work, we investigated the reactivity of Lewis acid sites in guaiacol catalytic pyrolysis by selectively poisoning the Brønsted acid sites using sodium ion exchange. We found that Lewis acid sites alone have only a negligible influence on the conversion[20]. On the other hand, Zheng et al.[28] found that nanocomposite materials such as $WO_3$-$TiO_2$-$Al_2O_3$ perform well towards deoxygenation in guaiacol, phenol, and creosol model compounds due to the combined effect of Lewis and Brønsted sites. Liao et al.[29] also observed high Brønsted acid site activity in the dealkylation of lignin-derived 4-*n*-propylphenol to phenol over HZSM5. Furthermore, Jiang et al.[26] pointed out the central role of Brønsted acid sites in the demethoxylation and dehydroxylation reactions in guaiacol catalytic pyrolysis. Because of their integral part in both fulvenone ketene and phenol formation, tuning the configuration of Brønsted acid sites is most likely to allow for effective selectivity control. Fulvenone formation requires intramolecular dehydration, while dehydroxylation to phenol only involves one of the hydroxyl functional groups in catechol. Therefore, increasing the Brønsted acid site density may increase the phenol yield by making it more likely that catechol will coordinate to two acid sites simultaneously, thereby isolating the hydroxyl groups. This can be achieved most easily by decreasing the Si/Al ratio in the zeolite.

We utilize sensitive, isomer-selective, and multiplexed operando photoelectron photoion coincidence (PEPICO) spectroscopy to test this hypothesis. PEPICO spectroscopy combines mass spectrometry and photoelectron spectroscopy to record photoion mass-selected threshold photoelectron spectra (ms-TPES) in complex reactive mixtures, which enables the detection and assignment of elusive and

reactive species[12,19,20]. Thanks to molecular beam sampling, the reactive species exiting the reactor are not quenched and detected directly. Besides quantifying the mole fractions of intermediates and products by operando PEPICO spectroscopy, we also performed experiments with thermogravimetric analysis coupled with mass spectrometer (TGA/MS) detection, quantified the acid sites of the different faujasite catalysts using IR spectroscopy, and measured $^{29}$Si MAS-NMR to determine the ratio of adjacent Brønsted acid sites. Our mechanism is further corroborated by quantum chemical calculations of the ketene reaction pathways. We observed a fivefold selectivity increase towards phenol, the desired CFP product, upon lowering the Si/Al ratio, driven by the suppression of fulvenone formation.

## Results

### Isomer-selective identification

*Operando* PEPICO spectroscopy was used to probe the effluent from a quartz microreactor packed with HZSM5(25), HFAU(40), HFAU(15), or HFAU(2.6) zeolite catalysts. Figure 2a shows the time-of-flight mass spectra measured at 10.5 eV photon energy and 530 °C reactor temperature. Under these conditions, guaiacol was almost fully converted (94–98% over all catalysts, see Supplementary Fig. 1). The *m/z* peaks were identified isomer-selectively based on their ms-TPES and literature reference spectra or Franck–Condon spectral modeling, as shown in Fig. 2b–d and Supplementary Fig. 2[30,31]. Besides the labeled products in Fig. 2a, we identified the methyl radical (*m/z* 15), 1-buten-3-yne (*m/z* 52), and 1,3-butadiene (*m/z* 54) (Fig. 2b and Supplementary Fig. 2). The observed products and intermediates agree with previous results on guaiacol and benzenediol conversion over HFAU and HZSM5 and with the mechanism in Fig. 1[18–20].

In Fig. 2a, cyclopentadiene (*m/z* 66) as well as fulvene and benzene (*m/z* 78) are the main products over HZSM5(25), while catechol (*m/z* 110) shows the highest intensity over HFAU(40) at comparable pressure, sample concentration, and reactor temperature. The Brønsted acid site density of HFAU(40) is 0.08 mmol/g, determined by pyridine-infrared (py-IR) spectroscopy. SEM images, py-IR experimental data, and BET surface area literature values of the faujasite materials used here are summarized in Supplementary Fig. 3 and Supplementary Table 1. With decreasing Si/Al ratio, the Brønsted acid site density increases to 0.18 mmol/g in HFAU(15), and phenol (*m/z* 94) becomes dominant in the product distribution, followed by methylphenol and anisole (both *m/z* 108). HFAU(2.6) shows more benzene (*m/z* 78) as well as less methylphenol and anisole (*m/z* 108). This suggests that the higher Brønsted acid site density (0.43 mmol/g) also enhances the

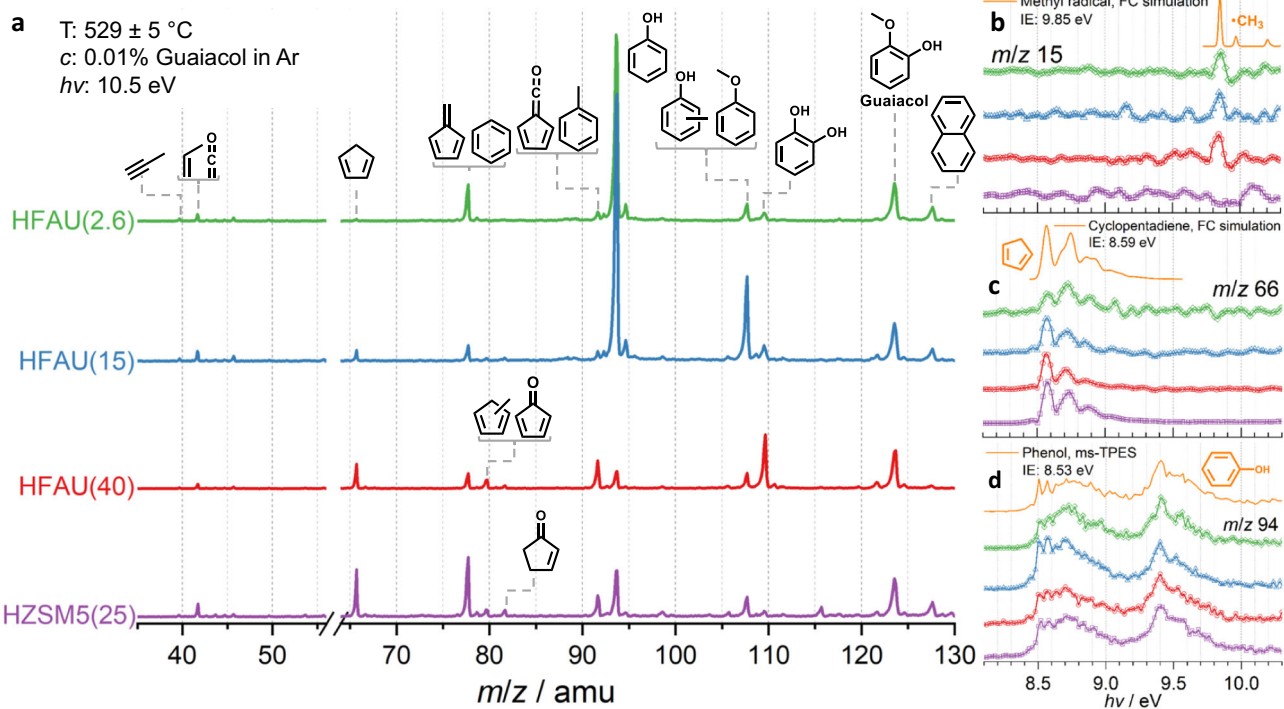

**Fig. 2 | Mass and photoion mass-selected photoelectron spectra of the catalytic fast pyrolysis products of guaiacol over zeolite catalysts. a** Time-of-flight mass spectra (ToF MS) recorded at 10.5 eV for 2 min. **b–d** ms-TPES of the desorbed methyl radicals ($m/z$ 15), cyclopentadiene ($m/z$ 66), and phenol ($m/z$ 94), respectively, shown together with simulations based on G4 ionization energy calculations and Franck–Condon (FC) simulated or experimental reference spectra.

proton-assisted dehydroxylation of phenol. Considering that methyl radicals can be hydrogenated to methane over Brønsted acid sites, methane production is likely promoted, and ring methylation is suppressed at low Si/Al ratios. Indeed, an increase in the methane signal is detected by TGA/MS when the Si/Al ratio is lowered (Supplementary Fig. 4b).

As shown in Fig. 1, guaiacol **1** CFP is initiated by demethylation and hydrogen transfer to yield catechol **2** (**1→2**)[20,23,24]. Catechol can undergo dehydroxylation to phenol (**2→3**, $m/z$ 94), dehydration to fulvenone (**2→4**, $m/z$ 92), or decarbonylation to 4-cyclopenten-1-one (**2→5**, $m/z$ 82). This mechanism is verified by varying the catalyst, the reaction temperature, and the reactant concentration (see Supplementary Fig. 5). At 440 °C, catechol ($m/z$ 110), methylphenol, and anisole ($m/z$ 108), as well as phenol ($m/z$ 94) are the main products on HZSM5(25). In contrast, the ToF MS over HFAU(40) is dominated by catechol at 434 °C. This suggests that fewer secondary reactions occur over HFAU(40), which is rationalized as follows: (i) the larger, 7.4 Å cage opening of the faujasite (FAU) framework leads to higher diffusivity, facilitating the departure of the primary products from the zeolite pores compared to the 5.5 Å cage opening of the pentasil MFI framework of HZSM5(25)[32]. Furthermore, (ii) a higher Si/Al ratio in HFAU(40) implies a lower Brønsted acid site density (Supplementary Fig. 3), thereby suppressing secondary reactions, such as catechol dehydroxylation or dehydration. When increasing the temperature further, the product distribution shifts to lower molecular weights and is dominated by fulvenone **4**, phenol **3** ($m/z$ 94), benzene **9** ($m/z$ 78), and cyclopentadiene **6** ($m/z$ 66) on HZSM5(25) and HFAU(40) (Supplementary Fig. 5). Finally, at 529 °C, both catalysts show similar product distributions (Fig. 2), which suggests that HZSM5 and HFAU share the same reaction mechanism (Fig. 1). Dehydration of catechol (**2→4**) was identified as the most important reaction over HZSM5 to yield the reactive fulvenone ketene **4**, which in turn is the main precursor to cyclopentadiene **6**[19,20].

Within the FAU framework, we also investigated HFAU(15) and HFAU(2.6) as catalysts. The abundance of Brønsted acid sites increases with decreasing Si/Al ratio (Supplementary Fig. 3). HFAU(15) and HFAU(2.6) show enhanced selectivity towards phenol and $m/z$ 108 (methylphenol and anisole, see Fig. 2a and Supplementary Fig. 5), while fulvenone and toluene ($m/z$ 92), benzene ($m/z$ 78), and cyclopentadiene ($m/z$ 66) yields are strongly suppressed. Together with the HFAU(40) and HZSM5(25) results under the same reaction conditions (Fig. 2a), this indicates that low Si/Al ratios increase the selectivity towards phenols, while fulvenone formation is largely suppressed over HFAU(15) and HFAU(2.6). The suppression of this reactive ketene and the associated change in the reaction mechanism is likely the key to promoting catechol dehydroxylation to phenol and responsible for the selectivity increase. To prove this hypothesis, we followed the products and intermediates based on their ms-TPES and obtained quantitative species profiles as a function of temperature, as depicted in Fig. 3.

**Tracing the reaction pathways**

To understand how faujasite catalysts with different Si/Al ratios affect the reaction mechanism, we follow the fate of catechol **2** and fulvenone **4**. On the one hand, catechol is the precursor of fulvenone **4**, which is prone to hydrogen transfer reactions and can decarbonylate to cyclopentadiene **6** (**4→6**, Fig. 1). Methylation by abundant surface methyl yields methyl cyclopentadienes **7**. After hydrogen loss of **7**, fulvene **8** is formed, which is a precursor to benzene **9** ($m/z$ 78)[33]. Thereafter, benzene can be further methylated to toluene **10**[18–20]. On the other hand and along a parallel pathway, phenol **3** can also be formed directly via dehydroxylation of catechol **2** (**2→3**, Fig. 1). Phenol further dehydroxylates to benzene (**3→9**), recently confirmed as a second pathway to benzene by studying the CFP of benzenediol isomers over HZSM5[19]. By comparing the benzene **9** vs. fulvene **8** ($m/z$ 78) as well as fulvenone **4** abundances as a function of the Si/Al ratio in HFAU, we can elucidate the phenol **3** formation mechanism and its

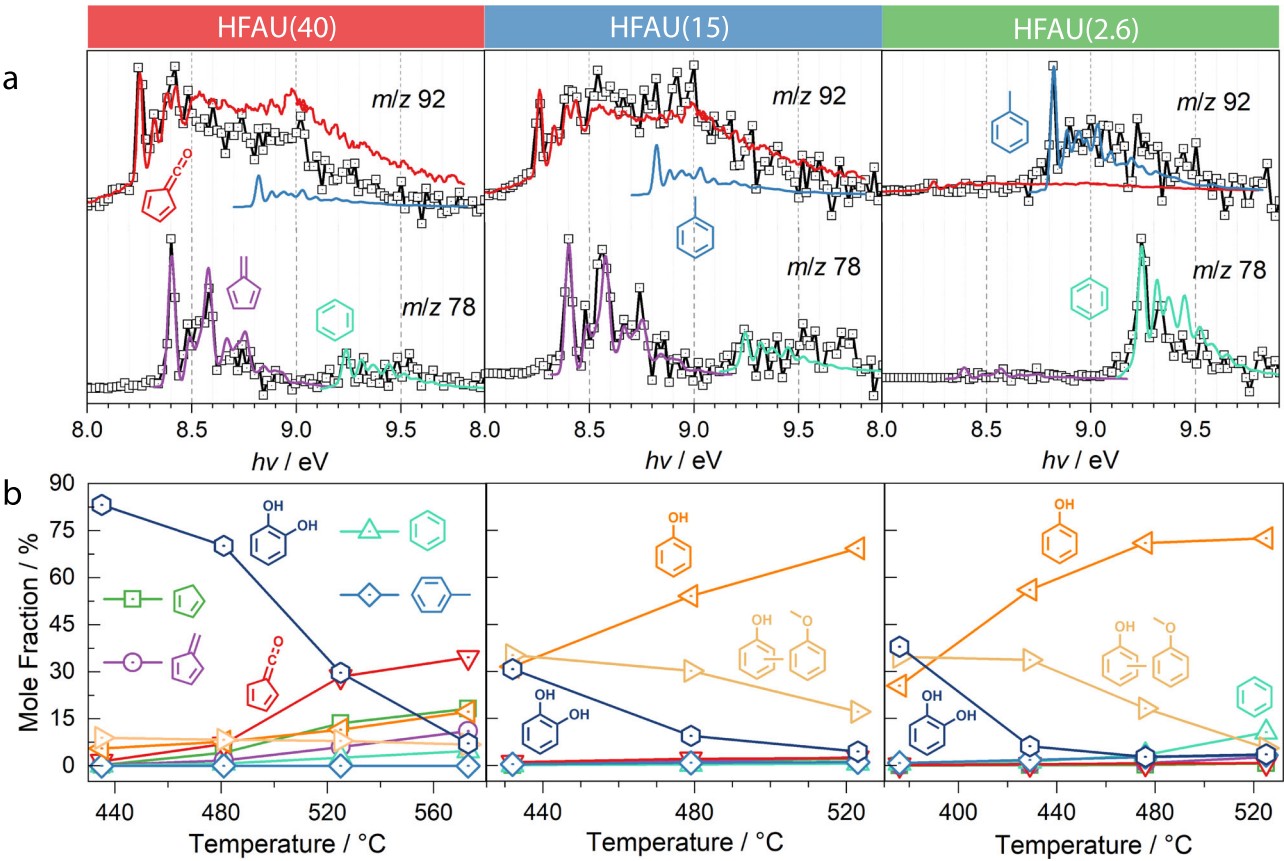

**Fig. 3 | Influence of different Si/Al ratios on guaiacol CFP products over HFAU.**
**a** Comparison of the experimental *m/z* 78 and 92 ms-TPES (squares) with reference spectra (colored lines) of spectral carriers. ms-TPES over HFAU(40), HFAU(15), and HFAU(2.6) were measured at 575, 531, and 528 °C, respectively. **b** Temperature-dependent mole fraction of the main products and intermediates in guaiacol CFP.

dependence on the Si/Al ratio of the catalyst. The details of our quantification strategy are given in Supplementary Note 1 and Supplementary Table 2.

Over HFAU(40), fulvenone **4** dominates the *m/z* 92 ms-TPES (Fig. 3a) and its mole fraction rises to 34% in the reaction stream at 575 °C (Fig. 3b) with no discernible toluene **10** contributions (blue trace, Fig. 3a)[34]. When lowering the Si/Al ratio, the mole fraction of fulvenone **4** plummets to less than 3% over HFAU(15) and it is virtually absent in the HFAU(2.6) experiment (Fig. 3b). The same tendency is found for the *m/z* 78 ($C_6H_6$) ms-TPES, where fulvene **8** (purple curve, Fig. 3a) vanishes and only benzene **9** is found at the lowest Si/Al ratio. Over HFAU(40), the catechol mole fraction (dark blue) decreases from 83 to 7% with increasing temperature, while fulvenone (red) gradually increases and becomes the main product above 525 °C at a mole fraction of 34% (Fig. 3b). Additionally, the mole fractions of cyclopentadiene (green), fulvene (purple) as well as benzene (light green) increase slightly with increasing temperature to reach 18%, 11%, and 5% at 573 °C, respectively. Over HFAU(15), the situation changes dramatically, and catechol, phenol (yellow), and methylphenol (light yellow) are the main products at low temperatures (432 °C). When increasing the temperature, phenol becomes dominant and its fractional abundance reaches 69% at 523 °C, while that of methylphenol and catechol decreases. Assuming that phenol is the precursor of methylphenol, the decreasing methylphenol signal suggests that methylation is suppressed at elevated temperatures. Moreover, toluene and benzene both exhibit mole fractions below 3% due to the high selectivity towards phenol. This effect is even more pronounced in HFAU(2.6), where phenol, methylphenol, and catechol have similar mole fractions as over HFAU(15), but already at lower reactor temperatures, which can be

explained by a higher reactivity at low Si/Al ratios. When comparing the mole fractions (selectivities) at fixed reactor temperature (ca. 480 °C) and varying the Brønsted acid site density in Supplementary Fig. 6 change in the reactivity is evident: While catechol still possess a large abundance in HFAU(40) (0.08 mmol/g) the reactivity of the catalyst and the selectivity towards phenol increases dramatically at 0.18 and 0.43 mmol/g Brønsted acid site concentration, respectively. In contrast, benzene (green curve, Fig. 3a) and toluene (*m/z* 92, blue curve) are the main products in the ms-TPES at 2.6 Si/Al ratio, while fulvenone and fulvene formation is almost fully suppressed, which signifies a change in the reaction mechanism. This means that phenol and benzene result exclusively from dehydroxylation of catechol (**1**→**2**→**3**→**9**, Fig. 1). Subsequent methylation of these initial products explains the formation of methyl phenol/anisole (*m/z* 108) and toluene (*m/z* 92), respectively. Due to the suppression of fulvenone formation, cyclopentadiene cannot be formed and, thus, its methylation to methyl cyclopentadiene **7** and reaction to fulvene **8** does not occur (**2**→**4**→**6**→**7**→**8**, Fig. 1).

Increasing the Brønsted acid site density suppresses fulvenone formation, which leads to a change in the reaction mechanism and, thus, to different product selectivities. The proposed mechanism by which the Brønsted acid site density influences the guaiacol CFP selectivity is illustrated in Fig. 4. Guaiacol catalytic fast pyrolysis is initiated by demethylation, producing catechol and a surface methyl species. The large abundance of the catechol primary product and the detection of methyl radicals desorbed from the catalyst surface evidence this reaction step[21–23]. Catechol can react further to phenol directly or to fulvenone on the catalyst. However, the density of Brønsted acid sites (Supplementary Fig. 6) influences the product selectivities fundamentally. When the Si/Al ratio is high, such as in

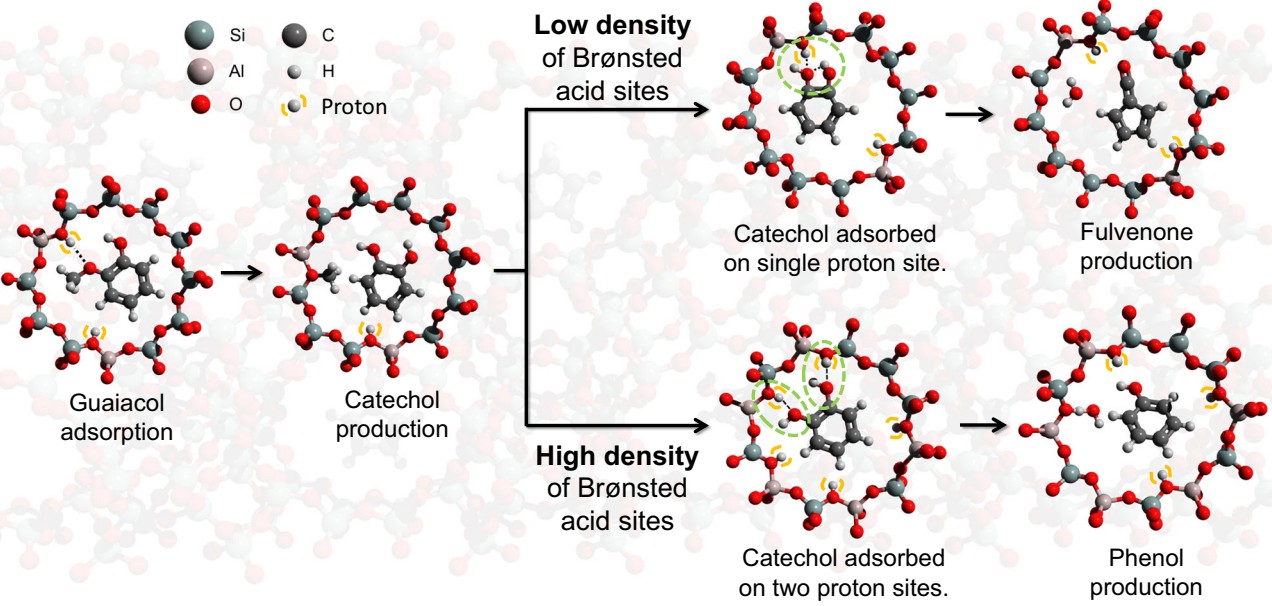

**Fig. 4 | Reaction pathways of guaiacol CFP.** The effect of Brønsted acid site density on the reaction mechanism.

HFAU(40), Brønsted acid sites are likely isolated. Catechol is, therefore, more likely bound to a single proton site (top branch in Fig. 4). A high Brønsted acid site density leads to an absorption configuration in which both hydroxyl groups are coordinated to the surface (lower branch in Fig. 4).

### Role of Brønsted acid sites

To confirm that Brønsted acid sites are indeed closer when their density is increased, we have performed $^{29}$Si MAS-NMR spectroscopy (Fig. 5a). The chemical shifts at −89, −96, −101, and −106 ppm were assigned to the $Q^4$(3Al), $Q^4$(2Al), $Q^4$(1Al) and $Q^4$(0Al) bands, respectively. These bands correlate with the concentration of the (Al−OH)$_3$−Si, (Al−OH)$_2$−Si, Al−OH−Si, and Si−O−Si sites in the faujasite, respectively. The HFAU(2.6) spectrum in green shows a ratio of ca. 16:10:3:1 of the $Q^4$(0Al), $Q^4$(1Al), $Q^4$(2Al), and $Q^4$(3Al) bands, respectively, which corresponds to a total of 13% adjacent Brønsted acid sites (Al−OH)$_2$−Si $Q^4$(2Al) and (Al−OH)$_3$−Si ($Q^4$(3Al)). In the HFAU(40) spectrum (red), on the other hand, bands of the adjacent site ($Q^4$(3Al), $Q^4$(2Al)) are quite small at <2.5% in sum. The HFAU(15) spectrum follows the same trend and has a total contribution of around 5% of adjacent Brønsted acid sites. This shows that, upon decreasing the Si/Al ratio in faujasites, not only the number of protons increases (see FTIR spectra in Supplementary Fig. 3) but also those in close proximity to a central silicon atom, which favors bidentate coordination of catechol as depicted in Fig. 4 (lower branch). Due to their higher acidity, these sites possess a much higher reactivity, favoring dihydroxylation as compared to dehydration and fulvenone formation. To understand the catalysis mechanism better, selected elementary reaction steps are examined next by quantum chemical calculations.

Let us consider the unimolecular and acid-catalyzed formation of the fulvenone ketene. In the first case, in the absence of a catalyst (see Fig. 5b), catechol dehydration (**I**) is endothermic by 129 kJ/mol and proceeds over a high activation barrier of 336 kJ/mol (**TS-I**) to yield the fulvenone ketene (**II**). The present G4 results are in excellent agreement with the ones by Khachatryan et al.[35], who found transition state and product energies of 345 and 134 kJ/mol at CBS-QB3 level of theory, respectively. Without Brønsted acid sites, e.g., on Na-USY[20], fulvenone can only be formed non-catalytically, by this unimolecular reaction. Condensation of surface methyl species in CFP reactions ultimately

leads to coke formation, associated not only with the deactivation of the catalyst, but also with hydrogen transfer reactions associated with PAH formation and growth[36,37]. A simple model reaction to describe this hydrocarbon pool mechanism[19,38] is the dehydrogenation of 1,2-dihydronaphthalene (dialin, $C_{10}H_{10}$) to naphthalene, which acts as a hydrogen source and promotes the formation of phenol and cyclopentadiene (c-$C_5H_6$) via fulvenone, respectively, according to R1 and R2 ($\Delta H_r$ at G4 level):

$$c\text{-}C_5H_4\text{=}C\text{=}O + C_{10}H_{10} \rightarrow C_{10}H_8 + C_6H_5OH \qquad \Delta H_r = -166 \text{ kJ/mol} \quad \text{(R1)}$$

$$c\text{-}C_5H_4\text{=}C\text{=}O + C_{10}H_{10} \rightarrow C_{10}H_8 + c\text{-}C_5H_6 + CO \qquad \Delta H_r = -65 \text{ kJ/mol} \quad \text{(R2)}$$

R1 and R2 are both exothermic reactions and can thus rationalize the observation of phenol **3**, cyclopentadiene **6**, methyl cyclopentadiene **7**, and fulvene **8** (see e.g., Figs. 2 and 3). This can be understood as the result of stepwise hydrogen transfer of fulvenone, intermediately yielding $C_6H_5O$ radical isomers, from which the phenoxy and the aldehyde-cyclopentadienyl radical further react, as detailed earlier[20]. While phenoxy is hydrogenated to afford phenol **3**, the cyclopentadienyl-type species readily decarbonylates and further reacts with a second hydrogen atom (from the hydrocarbon pool) to cyclopentadiene **6**.

To model the Brønsted acid-catalyzed reaction, we consider protonated catechol (**III** in Fig. 5b). Like in the unimolecular dehydration, protonated catechol (**III**) decomposes via a five-membered ring transition state (**TS-III**, 193 kJ/mol), which relaxes to a $H_3O\text{−}c\text{-}C_5H_4\text{=}C\text{=}O^+$ complex (**IV**) 114 kJ/mol relative to **III**. Complex **IV** can decomposes to fulvenone and $H_3O^+$ after an O−H cleavage reaction without a reverse barrier at 199 kJ/mol. Although the process appears to be more endothermic than the non-catalytic **I→TS-I→II**, the oxonium ion may return the proton to the catalyst to restore the neutral energetics seen before. At the same time, the reaction barrier is ca. 140 kJ/mol lower in energy, evidencing the catalytic function of the proton in fulvenone formation. Fulvenone is a weak H-bond acceptor (see Supplementary Fig. 7) and readily desorbs from the catalyst surface.

Two neighboring Brønsted acid sites may isolate the two OH groups in catechol and, thus, prevent unimolecular dehydration.

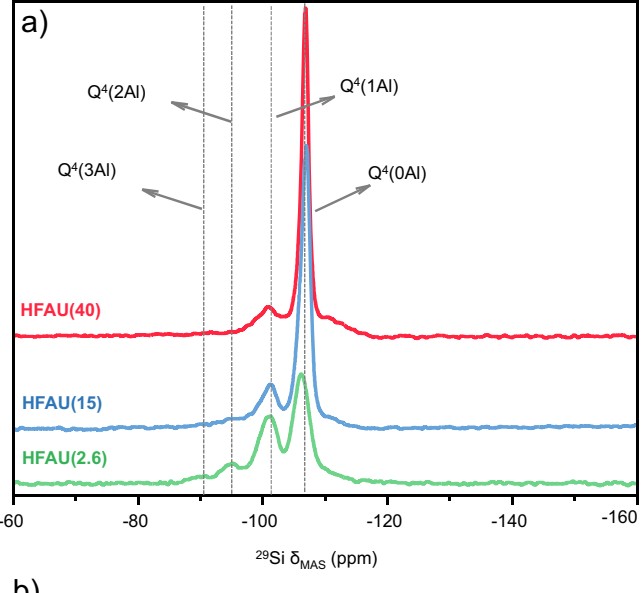

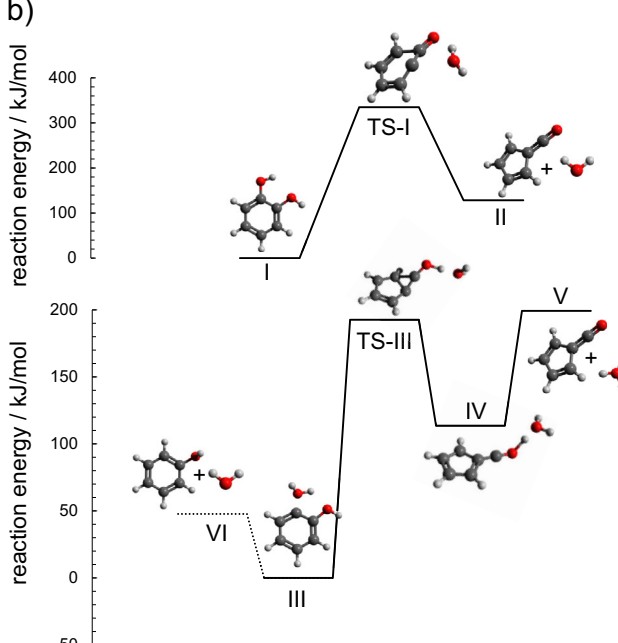

**Fig. 5 | ²⁹Si MAS-NMR spectra of HFAU(40-2.6) and reaction pathways. a** The fourfold increase in the $Q^4$(2Al) and $Q^4$(3Al) bands indicates an increasing amount of adjacent Brønsted acid sites when decreasing the Si/Al ratio. **b** G4 reaction pathways of catechol dehydration at unimolecular and acid-catalyzed conditions.

This promotes proton-assisted dehydroxylation to phenol instead. As discussed before and exemplified by dialin dehydrogenation, hydrogens are widely available through the hydrocarbon pool and can rapidly replenish the protons of the Brønsted acid sites. Hydrogen transfer from a close-lying Brønsted acid site to **III** and replenishment of the proton from the hydrocarbon pool is only endothermic by 48 kJ/mol (**III→VI**, Fig. 5b). This reaction will easily outcompete fulvenone formation if there are nearby Brønsted acid sites as well as a readily available hydrogen supply, provided by aromatic ring formation and condensation processes. The latter is evidenced by the factor of 2 faster deactivation of HFAU(2.6) vs. HFAU(40) and the detected increase in coke in the TGA analysis of the spent catalyst (see Supplementary Fig. 4a).

Transmethylation, a dominant process over HZSM5, is also suppressed at high Brønsted acid site densities. This is rationalized as

follows: surface-bound methyl usually reacts with surface intermediates to form methylcyclopentadiene, toluene, and xylene or is released into the gas phase to follow a radical-driven reaction pathway, as shown recently in methanol and methylchloride to olefin processes[11–13]. When Brønsted acid sites are abundant, e.g., in HFAU(2.6), surface-bound and gas-phase methyl is more likely hydrogenated to methane, as observed in TGA/MS (Supplementary Fig. 4b). The higher acidity is also associated with an increase in polycyclic aromatic hydrocarbons (PAHs) and coke formation[39], as evidenced by the threefold higher coke content of HFAU(2.6) compared to HFAU(40) in the TGA measurements (Supplementary Fig. 4b). Coke formation itself is the source of surface hydrogen, while it simultaneously decreases catalyst lifetime and represents the main catalyst deactivation pathway[40]. Mechanistic insights by operando tools can provide a way forward here, too, and can inspire mitigation strategies to involve an alternative hydrogen source to remove surface methyl species quickly while providing the necessary hydrogen for catechol dehydroxylation.

## Discussion

We investigated guaiacol catalytic pyrolysis using operando PEPICO spectroscopy and varied the Si/Al ratio in the zeolite catalyst to understand how acidity influences selectivities at the molecular level. We quantified products and intermediates, including fulvenone ketene, as a function of the Brønsted acid site density. TGA/MS allowed us to determine the amount of coke formed over faujasite catalysts. Pyridine FTIR measured the Brønsted acid site density, while ²⁹Si NMR spectroscopy confirmed the increasing number of neighboring acid sites as the acidity increased. By bidentate coordination of the hydroxyl groups in catechol, such acid site doublets and triplets are responsible for isolating them. This allowed us to directly associate increased acid site densities with a product selectivity increase towards phenol. Furthermore, we used quantum chemical calculations to show that catechol dehydroxylation to phenol is energetically preferred in the presence of abundant surface hydrogen, provided in our case by the formation of the hydrocarbon pool.

Mechanistically, this can be understood as follows: catalytic pyrolysis of guaiacol is initiated by demethylation, as evidenced by the detection of desorbed methyl radicals, to afford catechol. In the case of HZSM5(25) and HFAU(40), catechol dehydration dominates to yield fulvenone ketene, after which the reactive flux branches out to phenol via hydrogen transfer or cyclopentadiene via decarbonylation. This results in the unselective formation of methyl cyclopentadiene, the precursor of fulvene and benzene, along with phenol. Since fulvenone is a highly reactive intermediate, we hypothesized that selectivity control is achievable by suppressing fulvenone formation. Indeed, upon decreasing the Si/Al ratio in HFAU(15 and 2.6), the selectivity towards phenols increases fivefold. Mechanistically, the increased Brønsted acid site density in these catalysts suppresses catechol dehydration to fulvenone. Instead, we find that dehydroxylation is preferred to yield phenol, at a lower light-off temperature than over HZSM5(25) and HFAU(40). The absence of fulvenone, cyclopentadiene, methyl cyclopentadiene, and fulvene in the effluent proves that the side product benzene is also formed via dehydroxylation of phenol.

A molecular-level understanding of achieving high phenol yields in guaiacol CFP is desirable because guaiacol represents a common substructure in lignin and mimics some of the most abundant functional groups[41,42]. Moreover, the detection capabilities of PEPICO spectroscopy and the quantification approach of reactive species in combination with the product selectivity change as a function of the catalyst, are not limited to lignin catalytic pyrolysis or Brønsted acid sites. In fact, our approach relating reactive intermediate concentrations to changes in selectivity and conversion as a

function of the catalyst structure is broadly applicable to various heterogeneous catalytic processes, such as hydrogenation and syngas- or methanol-to-hydrocarbon reactions, where zeolites and other catalysts play a crucial role. Since ketenes and their surface analogs are important, especially during the formation of the first alkene in MTH, taking control of these species may have additional benefits for the overall selectivity. Operando PEPICO spectroscopy can aid catalyst fine-tuning to control product selectivities for targeted process optimization.

## Methods

### Operando photoelectron photoion coincidence spectroscopy (PEPICO)

Guaiacol catalytic fast pyrolysis was studied by photoelectron photoion coincidence spectroscopy at the vacuum ultraviolet (VUV) beamline of the Swiss Light Source (SLS) at the Paul Scherrer Institute[43]. The schematic of the operando PEPICO setup can be found elsewhere[19]. Glass wool soaked with guaiacol was placed in the sample container, of which the temperature was controlled by a water thermostat (Huber Minichiller) to control the guaiacol concentration in the gas phase[44]. The sample was carried by a 20 sccm argon (PanGas, 4.8) flow, controlled by a digital mass flow controller (MKS Instruments), and directed into a tubular quartz microreactor (2 mm inner diameter) at 0.5–1.0 bar pressure-packed with 9–11 mg powder catalyst. A cylindrical resistive heater connected to a DC power supply (Voltcraft) heated the reactor, and the reactor temperature was monitored by a type K thermocouple attached to the outside reactor wall. The effluent mixture including products, intermediates, and unconverted guaiacol desorbed from the catalyst surface left the reactor, and expanded into vacuum ($10^{-5}$ mbar). A beam was formed, in which reactive species are conserved thanks to the low collision frequency and the inert carrier gas. The central part of the expansion entered the ionization chamber ($10^{-6}$ mbar) via a 2 mm skimmer (Beam Dynamics). Bending magnet synchrotron radiation was collimated, dispersed by a monochromator with a 150 mm$^{-1}$ grating, and focused at a 200 µm exit slit at an energy resolution of $E/\Delta E \approx 1500$ in a differentially pumped noble gas filter. The gas filter was filled with 8.5 mbar of a mixture of Ar, Ne, and Kr over a path length of 10 cm to absorb high harmonic radiation of the grating. Monochromatic VUV radiation entered the ionization chamber and intersected the molecular beam at a 90° angle in the ionization region. The resulting photo-ions and -electrons were extracted in a constant 218 V cm$^{-1}$ electric field in opposite directions and were detected in delayed coincidence[43,45] using position-sensitive delay line detectors (RoentDek DLD40) to record time-of-flight mass spectra. By discriminating for close-to-zero kinetic energy electrons detected in coincidence with a cation in a single $m/z$ channel, photoion mass-selected threshold photoelectron spectra (ms-TPES) can be plotted to identify species by comparison with reference spectra or Franck–Condon simulations in conjunction with ionization energy calculations.

### Thermogravimetric analysis coupled with mass spectrometry (TGA/MS)

Guaiacol pyrolysis was investigated on a thermogravimetric analyzer (Netzsch STA 449 C) coupled with a mass spectrometer (Pfeiffer Vacuum OmniStar). The samples (20.4 mg for all the fresh mixtures of guaiacol with catalyst) were stabilized at 30 °C for 40 min under nitrogen (40 mL min$^{-1}$) and heated from 30 °C to 530 °C at a heating rate of 20 °C min$^{-1}$ under nitrogen (40 mL min$^{-1}$). Afterwards, the temperature was kept at 530 °C under nitrogen (40 mL min$^{-1}$) for 30 min. Finally, the samples were heated from 530 °C to 1000 °C at a heating rate of 20 °C min$^{-1}$ under a mixture of air and nitrogen (20 mL min$^{-1}$ each), and kept at 1000 °C under a mixture of air and nitrogen (20 mL min$^{-1}$ each) for 15 min.

### Materials

Guaiacol (Sigma–Aldrich, >97%) was used as received. Zeolite HZSM5(X) and HFAU(X), where X represents the Si/Al ratio, were bought from Zeolyst International. Prior to experiments, all zeolites were calcined at 550 °C at a heating rate of 2 °C min$^{-1}$ in static air and then kept at the final temperature for 6 h.

### Fourier-transform infrared (FTIR) spectroscopy

FTIR spectra of zeolite powders after pyridine adsorption were acquired on a Thermo Nicolet iS50 FTIR spectrometer equipped with a DTGS detector at an optical resolution of 4 cm$^{-1}$ by recording 128 scans. Prior to the measurements, ca. 20 mg of the zeolite sample was pressed in a self-supporting disc and activated in an IR transmission cell attached to a vacuum line at 450 °C for 4 h. For pyridine (Py) adsorption, the samples were exposed to 4 mbar of Py at 150 °C for 30 min for complete diffusion, followed by evacuation at 150 °C for 30 min. Difference spectra were obtained by subtracting the spectra of activated samples from those of the samples with adsorbed pyridine using OMNIC 9.3. The peaks at 1454 and 1544 cm$^{-1}$ were used to determine Lewis and Brønsted acid sites, respectively. For quantification, molar integral extinction coefficients of 2.22 and 1.67 cm µmol$^{-1}$ were used for Lewis and Brønsted acid sites, respectively[46].

### Solid-state NMR spectroscopy

Solid-state $^{29}$Si MAS-NMR spectra of the zeolite samples were recorded on a Bruker 400 MHz Ultra-Shield magnet and AVANCE III HD spectrometer at a resonance frequency of 79.51 MHz. The calcined zeolite powders (40–50 mg) were packed into 4 mm zirconia rotors. The packed rotors were spun at a spinning rate of 10 kHz with a 4 mm MAS probe for 10,000 scans to obtain the spectra. The chemical shifts of $^{27}$Si were referenced to octakis(trimethylsiloxy)silsesquioxane. All spectra were normalized to the sample weight. The Topspin 4.0.9 and Origin software packages were used for spectral processing and analysis.

### Scanning electron microscopy

For SEM, the powder form samples were carefully placed on a double-sided tape with the aluminum stub as the base. The observations were made at different magnifications using a Zeiss Leo 1530 scanning electron microscope.

### Computational

Quantum chemical computations were performed using the Gaussian 16 rev. A.03 suite of programs[47]. Geometry optimization and vibrational frequencies for Franck–Condon (FC) simulations were calculated by density functional theory at the B3LYP/6-31 G(d,p) and B3LYP/6-311G++(d,p) levels. Adiabatic ionization energies were calculated using the G4 composite method[48]. The stick spectra were convolved with a Gaussian function and compared to the experimental ms-TPES for isomer-specific assignment[18,19]. Reaction pathways were located using constrained geometry scans at B3LYP/6-311+G(d,p) level of theory and by synchronous transit-guided quasi-Newton calculations. The energetics were further refined by G4 composite method calculations.

## Data availability

The raw and derived data generated in this study have been deposited in the PSI Public Data Repository and can be accessed here https://doi.org/10.16907/2ea9f931-2939-4b55-a213-8ffa0b47a624.

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

## Acknowledgements

The measurements have been performed at the VUV (x04db) beamline at the Swiss Light Source, Paul Scherrer Institute in Switzerland. This work was funded by the Swiss National Science Foundation (SNSF, 200021_178952). The authors thank Patrick Ascher for technical assistance. A.P.-U. thanks the Ministry of Science, Technology, and Telecommunications of Costa Rica, and the Costa Rica Institute of Technology for their support.

## Author contributions

Z.P. performed experiments, analyzed data, contributed materials/analysis tools, and wrote the paper. A.P.-U. performed experiments and analyzed data. S.R.B. performed experiments and analyzed data. A.B. discussed experiments and wrote the paper. X.W. discussion of experiments. Z.Z. discussion of experiments. J.A.v.B. discussion of experiments. P.H. conceived and designed the experiments, discussion of experiments, wrote the paper, acquired funding, and project management. All authors commented on the paper.

## Competing interests

The authors declare no competing interests.
