## [Peer review File · Nature Communications]

REVIEWER COMMENTS

Reviewer #1 (Remarks to the Author):

The manuscript reports that tuning the zeolite acidity enables selectivity control by suppressing ketene formation. The authors declare that the Brønsted acid site density of the zeolite catalyst efficiently controls the guaiacol catalytic pyrolysis mechanism. At high Brønsted acid site density, fulvenone formation is inhibited and it is responsible for the five-fold phenol selectivity increase. The finding is very interesting and is important for the selectivity control in catalytic fast pyrolysis of lignin via rational catalyst design. The manuscript can be accepted after minor revisions.

1. The title of this manuscript should be revised, the selectivity control for specific reactions (e.g., catalytic fast pyrolysis of lignin or its model compound) should be mentioned.
2. The authors conclude that the Brønsted acid site density of catalyst is key to control the product selectivity in catalytic fast pyrolysis of guaiacol. If possible, the authors could present a quantitative relationship between the product selectivity and Brønsted acid site density.
3. Previous work demonstrated that the other solid acid catalyst, such as WO₃-TiO₂-Al₂O₃ could also control the product selectivity during catalytic fast pyrolysis of lignin. Are the authors' findings also valid for these non-zeolite catalysts? If possible, please discuss it.
4. Thermogravimetric analysis coupled with mass spectrometry (TGA/MS) is used to investigate the formation of CH₄ and coke in this study, However, the heating rate of TGA is much lower than the actual operating conditions of catalytic fast pyrolysis. Please clarify it.
5. The pore structure of the zeolite catalysts may also manipulate the reaction intermediates, it is better to provide a characterization of the pore structure of the catalysts used in this study.

Reviewer #2 (Remarks to the Author):

Overall the paper is very well done and I think would be a great fit for the journal. The paper is well written and uses a variety of techniques to explain the reaction chemistry involve in guaiacol conversion. I enjoyed reading about how they used a combination of photoelectron spectroscopy, DFT and kinetic analysis to elucidate the effects of acid sites on the reaction chemistry. I only have issues with two of their statements that could probably be slightly revised. The authors state in the last paragraph, "guaiacol is one of the most abundant early decomposition products of lignin." No reference is made for this statement. However, lignin does not form high yields of guaiacol. Lignin really form lignin oligomers (see Green Chemistry 2017, 19, 1378-1389 and similar references). To really convert lignin to something useful you need to study lignin oligomers not lignin monomers, which are only produced in very low yields. My other concern is some of the statements the authors make about rational catalyst design. The authors state, "we show how mechanistic details can guide rational catalyst design to induce a mechanism change and steer the reactive flux towards a preferred final product." I do not see evidence of catalyst design in this article. Instead I think this paper is a very well done detailed mechanistic study that clearly explains the reaction chemistry. The authors are testing some fairly conventional zeolites and it is not clear to me that these studies will aid in the design of a new type of zeolite for guaiacol conversion.

Reviewer #3 (Remarks to the Author):

Pan et al. investigated the catalytic pyrolysis of guaiacol by a set of complementary techniques, the most importantly the operando PEPICO spectroscopy. The corresponding research group recently delivered numerous good publications using PEPICO, which provided crucial mechanistic information, that was elusive before. In this work, they have correlated their mechanistic information with respect to the physicochemical properties of zeolites too, as they have used different zeolites with different SARs. This kind of project design could shed more definitive light on the zeolite catalysis and provide a molecular-level understanding of how acidity influences the mechanism and, eventually, the chemical process. This project was designed meticulously, and the experimental and analytic parts of this work are extremely high level. I believe this work poses sufficient novelty to be published in Nature Communication. However, I think the authors need to

clarify or explain some mechanistic interpretations, as listed below.

1) Firstly, I am not convinced by the term "hydrogenation" used throughout the manuscript. To me, most are hydrogen transfer or simple radical combinations of H-radical (e.g., methyl radical to methane). Authors should take appropriate pre-caution while describing each reaction step related to this matter. Abundant surface hydrogen could not always be correlated to hydrogenation.

2) The story currently entirely relies on Bronsted acid site catalysis. Authors barely touched the impact of Lewis acid sites (LAS), although they have experimentally quantified. Herein, I am not sure if LASs have any impact (or not!!). Nevertheless, this matter should be addressed because such alkylation/dealkylation could also be governed by LAS, especially at that reaction temperature.

3) I liked the correlation between ketene's role and the acidity of zeolite. To me, it is the most significant selling point of the work, which could be extrapolated to other zeolite-catalyzed hydrocarbon conversion processes as well. Based on my understanding, ketene plays a descriptor role within a certain concentration range; beyond that, it works as a spectator and promotes coke formations. Because ketene is essential anyway to get the desired product. Is my understanding correct? Nevertheless, I would advise authors to write some direct statements on this matter, which might be helpful for other zeolite researchers who are not familiar so much with this reaction or the technique itself.

Response to reviewer

Reviewer #1 (Remarks to the Author):

The manuscript reports that tuning the zeolite acidity enables selectivity control by suppressing ketene formation. The authors declare that the Brønsted acid site density of the zeolite catalyst efficiently controls the guaiacol catalytic pyrolysis mechanism. At high Brønsted acid site density, fulvenone formation is inhibited and it is responsible for the five-fold phenol selectivity increase. The finding is very interesting and is important for the selectivity control in catalytic fast pyrolysis of lignin via rational catalyst design. The manuscript can be accepted after minor revisions.

Answer:

We thank the referee for the kind words.

1. The title of this manuscript should be revised, the selectivity control for specific reactions (e.g., catalytic fast pyrolysis of lignin or its model compound) should be mentioned.

Answer/Action:

We revised the title accordingly: "Tuning the Zeolite Acidity Enables Selectivity Control by Suppressing Ketene Formation in Lignin Catalytic Pyrolysis."

2. The authors conclude that the Brønsted acid site density of catalyst is key to control the product selectivity in catalytic fast pyrolysis of guaiacol. If possible, the authors could present a quantitative relationship between the product selectivity and Brønsted acid site density.

Answer:

The quantitative relationship between the product selectivity and Brønsted acid site density at ca. 480 °C is presented below. The Catechol and fulvenone mole fractions remain high when the Brønsted acid site density is low. Upon increasing the acid site density in HFAU(15 and 2.6), the phenol selectivity dramatically increases, while catechol is consumed. These results indicate that Brønsted acid sites promote the catechol decomposition to phenol rather than fulvenone formation.

Actions:

We discuss the quantitative relationship in the manuscript on page 10: "When comparing the mole fractions (selectivities) at fixed reactor temperature (ca. 480 °C) and varying the Brønsted acid site density in Supplementary Fig. 6, a change in the reactivity is evident: While catechol still possess a large abundance in HFAU(40) (0.08 mmol g⁻¹) the reactivity of the catalyst and the selectivity towards phenol increases dramatically at 0.18 and 0.43 mmol g⁻¹ Brønsted acid site concentration, respectively.

In contrast, benzene (green curve, Fig. 3a) and toluene (m/z 92, blue curve) are the main products in the ms-TPES at 2.6 Si/Al ratio, while fulvenone and fulvene formation is almost fully suppressed, which signifies a change in the reaction mechanism.”

Supplementary Fig. 6 | Mole fraction (selectivities) of the main products and intermediates as a function of the density of HFAU Brønsted acid site concentration in guaiacol CFP at ca. 480 °C.

3. Previous work demonstrated that the other solid acid catalyst, such as WO₃-TiO₂-Al₂O₃ could also control the product selectivity during catalytic fast pyrolysis of lignin. Are the authors' findings also valid for these non-zeolite catalysts? If possible, please discuss it.

Answer:

Thank you very much for pointing this out. We assume the referee is referring to the work by Zheng et al. (DOI:10.1016/j.isci.2019.100814). In this work, demethoxylation to yield methanol and DME seems to be an important channel for the deoxygenation. In our results, neither methanol nor DME could be observed, which deems these channels to be minor. On the other hand, without investigating nanocomposites like WO₃-TiO₂-Al₂O₃ we cannot rule out dehydroxylation mechanisms similar to our results in non-zeolite catalysts.

Actions:

We have added one sentence on the suggested catalysts to the manuscript, which now reads: “On the other hand, Zheng et al. found that nanocomposite materials such as WO₃-TiO₂-Al₂O₃ perform well towards deoxygenation in guaiacol, phenol, and creosol model compounds due to the combined effect of Lewis and Brønsted acid sites.”

4. Thermogravimetric analysis coupled with mass spectrometry (TGA/MS) is used to investigate the formation of CH₄ and coke in this study. However, the

heating rate of TGA is much lower than the actual operating conditions of catalytic fast pyrolysis. Please clarify it.

Answer:

It is correct that using TGA-MS the experiments were limited regarding the heating rates that could be applied. In our experiments, the samples were heated from 30 °C to 530 °C at a rate of 20 °C min⁻¹ and then kept at 530 °C for 30 min (both thermal steps carried out under a nitrogen atmosphere). To reduce the influence of the slow heating rate in our results, the mass spectrometer continuously collected data (all signals were summed up) during the two thermal steps mentioned (i.e., for 55 min), with most of the time at a temperature around which all the catalysts tested presented a similar conversion (as shown in Supplementary Fig. 1).

Actions: We have adapted the caption of Supplementary Fig. 4, which now reads: “Guaiacol pyrolysis studied by thermogravimetric analysis coupled with mass spectrometry (TGA/MS). **a**, Mass loss as a function of temperature. **b**, Methane and coke yields as function of the Si/Al ratio. To reduce the influence of the slow heating rate in our results, the mass spectrometer continuously collected data (all signals were summed up) during the two thermal steps mentioned (i.e., for 55 min), with most of the time at a temperature around which all the catalysts tested presented a similar conversion (as shown in Supplementary Fig. 1)”

5. The pore structure of the zeolite catalysts may also manipulate the reaction intermediates, it is better to provide a characterization of the pore structure of the catalysts used in this study.

Answer:

The referee is right, that the pore structure and active sites both play important roles in the reactions. In this work, we mainly focus on the influence of active sites. To ensure a consistent pore structure, we used the same type zeolite FAU. Table 1 shows the pore volume of HFAU with different Si/Al ratios, and it indicates that all catalysts have similar micropore volume (within the error limits, i.e. ±20%) while the mesopore volume increases with the Si/Al ratio. According to the SEM images (*Fig 2*), HFAU(15 and 40) possess additional pores on the surface, which may explain the increase of mesoporous volume.

Supplementary Table 1 | Textural properties of HFAU catalysts

	V _{micro} (cm ³ /g)	V _{meso} (cm ³ /g)	V _{total} (cm ³ /g)	Ref.
HFAU(2.6)	0.27	0.02	0.30	1
HFAU(15)	0.30	0.13	0.44	
HFAU(40)	0.21	0.26	0.46	2

¹ACS Catal. 2015, 5, 754–768. ²Materials 2020, 13, 4013.

Supplementary Fig. 3 | Lower trace SEM images of HFAU catalysts. a) HFAU(2.6), b) HFAU(15), c) HFAU(40).

Actions:

We have added the SEM figures as well as the table in the SI (Supplementary Fig. 3, Supplementary Table 1), and have added a paragraph on page 4 of the manuscript, which now reads: "The Brønsted acid sites density of HFAU(40) is 0.08 mmol/g, determined by pyridine-infrared (py-IR) spectroscopy. SEM images, py-IR experimental data and BET surface area literature values of the faujasite materials used here are summarized in Supplementary Fig. 3 and Supplementary Table 1."

Reviewer #2 (Remarks to the Author):

Overall the paper is very well done and I think would be a great fit for the journal. The paper is well written and uses a variety of techniques to explain the reaction chemistry involve in guaiacol conversion. I enjoyed reading about how they used a combination of photoelectron spectroscopy, DFT and kinetic analysis to elucidate the effects of acid sites on the reaction chemistry. I only have issues with two of their statements that could probably be slightly revised.

Answer:

We are grateful to the referee for the encouragement.

1. The authors state in the last paragraph, "guaiacol is one of the most abundant early decomposition products of lignin." No reference is made for this statement. However, lignin does not form high yields of guaiacol. Lignin really form lignin oligomers (see Green Chemistry 2017, 19, 1378-1389 and similar references). To really convert lignin to something useful you need to study lignin oligomers not lignin monomers, which are only produced in very low yields.

Answer/Action:

This sentence has been adopted: "A molecular-level understanding of achieving high phenol yields in guaiacol CFP is desirable, because guaiacol represents a common substructure in lignin and mimics some of the most abundant functional groups. (*Current Opinion in Biotechnology* 2019, 56:240–249; *Green Chemistry* 2017, 19, 1378-1389)"

We agree with the opinion of the referee that oligomers should be the next step to unveil lignin CFP mechanisms, which will provide more insights into lignin mechanisms. However, the reaction mechanism of oligomers is very complex and needs the insights from smaller model compounds, too. According to the bottom-to-up strategies, we started to understand the most basic model compounds, such as benzenediol and guaiacol, firstly and based on that, we are looking at more complicated model compounds, such as eugenol and some dimers containing β -O-4 linkage. As the referee suggested, oligomers will be the model compounds of choice once we understand the reaction mechanisms of smaller model compounds.

2. My other concern is some of the statements the authors make about rational catalyst design. The authors state, "we show how mechanistic details can guide rational catalyst design to induce a mechanism change and steer the reactive flux towards a preferred final product." I do not see evidence of catalyst design in this article. Instead I think this paper is a very well done detailed mechanistic study that clearly explains the reaction chemistry. The authors are testing some fairly conventional zeolites and it is not clear to me that these studies will aid in the design of a new type of zeolite for guaiacol conversion.

Answer:

The referee is right that we did not design a catalyst in this manuscript. We used different methods, especially operando PEPICO, to trace reactive intermediates over different catalysts, which helps us to understand the influence of molecular-level catalytic mechanisms and build up the relationship between the catalysts' active sites and product selectivity. Based on that, we propose the option to control the product selectivity by adjusting/designing catalysts.

Actions:

The statement about catalyst design has been revised and now reads (p3): "In this contribution, we show how unveiling mechanistic details can help to steer the reactive flux towards a preferred final product. The strategies introduced here may guide rational catalyst design to induce a mechanism change in other catalytic systems, too"

Reviewer #3 (Remarks to the Author):

Pan et al. investigated the catalytic pyrolysis of guaiacol by a set of complementary techniques, the most importantly the operando PEPICO spectroscopy. The corresponding research group recently delivered numerous good publications using PEPICO, which provided crucial mechanistic information, that was elusive before. In this work, they have correlated their mechanistic information with respect to the physicochemical properties of zeolites too, as they have used different zeolites with different SARs. This kind of project design could shed more definitive light on the zeolite catalysis and provide a molecular-level understanding of how acidity influences the mechanism and, eventually, the chemical process. This project was designed

meticulously, and the experimental and analytic parts of this work are extremely high level. I believe this work poses sufficient novelty to be published in Nature Communication.

Answer:

We are grateful for the positive feedback from the referee.

However, I think the authors need to clarify or explain some mechanistic interpretations, as listed below.

1) Firstly, I am not convinced by the term "hydrogenation" used throughout the manuscript. To me, most are hydrogen transfer or simple radical combinations of H-radical (e.g., methyl radical to methane). Authors should take appropriate pre-caution while describing each reaction step related to this matter. Abundant surface hydrogen could not always be correlated to hydrogenation.

Answer:

We thank the referee for this suggestion and corrected the term hydrogenation throughout the manuscript. We now distinguish between hydrogen radical addition, hydrogen transfer, and (de)hydrogenation reactions.

2) The story currently entirely relies on Brønsted acid site catalysis. Authors barely touched the impact of Lewis acid sites (LAS), although they have experimentally quantified. Herein, I am not sure if LASs have any impact (or not!!). Nevertheless, this matter should be addressed because such alkylation/dealkylation could also be governed by LAS, especially at that reaction temperature.

Answer:

This is an important point, and we are glad that the referee brought this up. To distinguish the role of Brønsted and Lewis acid sites, we would like to discuss our work where we compared the reactivity of H-USY and Na-USY during the guaiacol catalytic pyrolysis (Nat. Commun. | 8:15946 | DOI: 10.1038/ncomms15946). In the absence of Brønsted acid sites (Na-USY) the reactivity and the conversion were low. This may indicate that Lewis acid sites may only play a subordinate role.

Actions:

In the manuscript, a related interpretation is presented on page 4: "Compared to Lewis acid sites, Brønsted acid sites represent the main active site in catalytic pyrolysis over zeolites.²⁵⁻²⁷ In our previous work, we investigated the reactivity of Lewis acid sites in guaiacol catalytic pyrolysis by selectively decreasing the concentration of Brønsted acid sites using sodium ion exchange. We found that Lewis acid sites alone have only a negligible influence on the conversion.²⁰"

3) I liked the correlation between ketene's role and the acidity of zeolite. To me, it is the most significant selling point of the work, which could be extrapolated to other zeolite-catalyzed hydrocarbon conversion processes as well. Based on my understanding, ketene plays a descriptor role within a certain concentration range; beyond that, it works as a spectator and promotes coke formations. Because ketene is essential anyway to get the desired product. Is my understanding correct? Nevertheless, I would advise authors to write some direct statements on this matter, which might be helpful for other zeolite researchers who are not familiar so much with this reaction or the technique itself.

Answer:

The referee is right that our findings could be translated to other zeolite catalyzed reactions. Fulvenone is a viable descriptor of this reaction as during its presence the reaction is less selective due to its high reactivity. Its suppression by means of increasing the Brønsted acid sites density, on the other hand, tames the reaction making it more controllable to yield more phenol.

Actions:

We emphasized this point especially on p.15: "Since fulvenone is a highly reactive intermediate, we hypothesized that selectivity control is achievable by suppressing fulvenone formation." and on p.16 "In fact, our approach relating the reactive intermediates' concentrations to changes in selectivity and conversion as a function of the catalyst structure is broadly applicable to various heterogeneous catalytic processes, such as hydrogenation and syngas- or methanol-to-hydrocarbon (MTH) reactions, where zeolites and other catalysts play a crucial role. Since ketenes and their surface analogs are important, especially during the formation of the first alkene in MTH, taking control of these species may have additional benefits for the overall selectivity"

REVIEWERS' COMMENTS

Reviewer #1 (Remarks to the Author):

The manuscript can be accepted in its current form.

Reviewer #2 (Remarks to the Author):

The authors have adequately responded to all of my concerns. I recommend the paper for publication.

Reviewer #3 (Remarks to the Author):

The authors have addressed all comments of reviewers. I have no other concerns. Hence, the manuscript can be accepted as it is.